# Are long-term growth responses to elevated $p$CO$_2$ sex-specific in fish?

Christopher S. Murray[ORCID]$^{1}$*, Hannes Baumann$^{2}$

**1** Washington Ocean Acidification Center, School of Marine and Environmental Affairs, University of Washington, Seattle, WA, United States of America, **2** Department of Marine Sciences, University of Connecticut, Groton, CT, United States of America

* csm187@uw.edu

**Data Availability Statement:** All relevant data are within the manuscript and its Supporting Information files.

**Funding:** This work funded by the National Science Foundation (https://www.nsf.gov/) OCE #1536165, awarded to Hannes Baumann. The funders had no

## Abstract

Whether marine fish will grow differently in future high $p$CO$_2$ environments remains surprisingly uncertain. Long-term and whole-life cycle effects are particularly unknown, because such experiments are logistically challenging, space demanding, exclude long-lived species, and require controlled, restricted feeding regimes—otherwise increased consumption could mask potential growth effects. Here, we report on repeated, long-term, food-controlled experiments to rear large populations (>4,000 individuals total) of the experimental model and ecologically important forage fish *Menidia menidia* (Atlantic silverside) under contrasting temperature (17˚, 24˚, and 28˚C) and $p$CO$_2$ conditions (450 vs. ~2,200 µatm) from fertilization to ~ a third of this annual species' life span. Quantile analyses of trait distributions showed mostly negative effects of high $p$CO$_2$ on long-term growth. At 17˚C and 28˚C, but not at 24˚C, high $p$CO$_2$ fish were significantly shorter [17˚C: -5 to -9%; 28˚C: -3%] and weighed less [17˚C: -6 to -18%; 28˚C: -8%] compared to ambient $p$CO$_2$ fish. Reductions in fish weight were smaller than in length, which is why high $p$CO$_2$ fish at 17˚C consistently exhibited a higher Fulton's $k$ (weight/length ratio). Notably, it took more than 100 days of rearing for statistically significant length differences to emerge between treatment populations, showing that cumulative, long-term CO$_2$ effects could exist elsewhere but are easily missed by short experiments. Long-term rearing had another benefit: it allowed sexing the surviving fish, thereby enabling rare sex-specific analyses of trait distributions under contrasting CO$_2$ environments. We found that female silversides grew faster than males, but there was no interaction between CO$_2$ and sex, indicating that males and females were similarly affected by high $p$CO$_2$. Because Atlantic silversides are known to exhibit temperature-dependent sex determination, we also analyzed sex ratios, revealing no evidence for CO$_2$-dependent sex determination in this species.

## Introduction

Human activities are rapidly increasing atmospheric and therefore surface ocean carbon dioxide (CO$_2$) [1]. With the unmitigated production of anthropogenic CO$_2$ (i.e., RCP8.5 emissions scenario) these levels could eclipse 2,000 ppmv within the next 300 years [2]. The rapid

role in study design, data collection and analysis, decision to publish, or preparation of the manuscript.

**Competing interests:** The authors have declared that no competing interests exist.

progression of modern ocean acidification (OA) may challenge physiological tolerance limits of many marine ectotherms [3–5]. In marine fish, responses to future $p\text{CO}_2$ conditions have been complex. Experiments have demonstrated a range of positive, neutral, and negative impacts to survival, development, and behavior [6–8]. Potential effects on fish growth are of particular interest, given the established link between individual growth and fitness [9–11] and the theoretical expectation that hypercapnia demands increased energetic allocations to acid-base homeostasis while reducing hemoglobin-oxygen binding efficiency [12–14], thus decreasing growth. For juvenile and adult fish, however, such metabolic tradeoffs have largely proved undetectable [15]. By contrast, laboratory studies on fish early life stages with still developing acid-base proficiency have demonstrated reduced growth in some but not most cases [16–20]. Meta-analyses across fish species and life-stages have therefore concluded that there are no consistent growth effects of high $p\text{CO}_2$ [6, 21].

While this may underscore the general $\text{CO}_2$ tolerance of fish as highly mobile vertebrates, the variability in reported growth responses may also partially be due to methodological constraints [22]. First, OA experiments on fish have mainly studied short-term responses to high $p\text{CO}_2$ within a single life-stage, thereby encompassing just a small fraction of a species' lifespan. Elevated $p\text{CO}_2$ conditions likely elicit a range of acclimation responses, including the differential expression of key regulatory enzymes [23, 24] and the maintenance of elevated bicarbonate in extra-cellular fluids [25]. While the energetic cost of these pathways may be too small to detect on short time scales [14, 26, 27], few studies have quantified how continuous energetic costs of $\text{CO}_2$ acclimation may accumulate over time and thus perhaps result in detectable growth effects at later life stages [28, 29]. Second, most OA studies on fish have employed relatively low levels of replication and small sample sizes, which allows detecting major effects but limits statistical power to detect other, potentially more subtle shifts in response traits [30]. Third, laboratory OA studies often provide excess food rations to fish offspring to avoid the confounding effects of uneven food supply. While logistically practical, this approach may enable fish to increase consumption to match energetic requirements and thus mask negative growth effects. To date, most studies exploring a link between ration level and $\text{CO}_2$ sensitivity have reported neutral responses [31–33], but negative interactions have also been documented [34]. In short, the emergent consensus that high $p\text{CO}_2$ environments do not affect fish growth may not be as robust as the current body of empirical data suggests. Moreover, temperature introduces further complexity when disentangling how $\text{CO}_2$ affects fish metabolism. Efficient acclimation to hypercapnia may depend on thermal conditions [35] but here again a consensus regarding interactive effects of $\text{CO}_2$ and temperature has remained elusive [36].

A so far underexamined aspect of OA is the potential for sex-specific physiological impacts [37]. Because sexes face different energetic tradeoffs associated with growth and reproduction [38, 39] the cost of $\text{CO}_2$ acclimation could disproportionately affect one sex over the other. Female fish that have the added energetic cost of maturing oocytes may incur a larger growth deficit when continuously exposed to OA conditions. Given the positive relationship between female body size and reproductive success [38], data on sex-specific $\text{CO}_2$ effects are critically needed [37]. Furthermore, a reduction in ocean pH could influence the sex ratios of species that exhibit environmental sex determination. While temperature is the most common abiotic cue that controls environmental sex determination in fish [40], in some freshwater teleosts, exposure to low pH conditions can result in a higher proportion of males in the population [41–43]. In the Atlantic silverside (*Menidia menidia)*, exposure to warm conditions (>17˚C) during early larval development (between 8–21 mm total length) has a masculinizing effect [44] because warm temperatures suppress the expression of the feminizing enzyme aromatase which promotes the development of testes [45, 46]. Warm temperatures are typically

correlated with more acidic conditions in productive nearshore environments [47], hence, temperature-dependent sex determination (TSD) in silversides could also be sensitive to pH. This hypothesis has so far remained untested.

Over the course of three years, we repeatedly reared large experimental populations of Atlantic silversides (>4,000 individuals total) from fertilization to more than a third of their lifespan under future (~2,200 μatm) versus present-day (~450 μatm) $pCO_2$ conditions and three temperatures (17˚, 24˚, and 28˚C). We administered non-excess feeding conditions by incrementally adjusting food rations based on the number and calculated biomass of individuals in each rearing tank. Sub-samples across developmental stages allowed examining if and when growth differences would manifest. Additionally, large random subsets of juveniles were sexed to determine sex ratios and potential sex-specific effects of high $pCO_2$ environments. We hypothesized that long-term exposure to acidified conditions would cause small but continuous reallocation of energetic resources away from growth, resulting in smaller fish of lower condition. We further predicted that sub-optimal rearing temperatures (17˚ and 28˚C) would exacerbate deleterious $CO_2$ effects. Last, we predicted that acidified conditions incur greater growth deficits in females than males and produce more male biased populations.

## Methods

### Experimental $CO_2$ and temperature conditions

Experiments were conducted in 700-L circular tanks. Two contrasting $pCO_2$ conditions were tested; ambient (~450 μatm $pCO_2$, $pH_{NIST}$ = ~8.05) versus high $pCO_2$ corresponding to the upper-end projection for the next 280 years under RCP8.5 [~2,200 μatm $pCO_2$, $pH_{NIST}$ = ~7.50, 2]. The two $pCO_2$ levels were crossed with three temperature conditions: 17˚, 24˚, 28˚C. The lower two temperatures (17˚ and 24˚C) encompass the thermal experience of silversides during their spawning season at this latitude [48], with ~24˚C considered to be the species' optimal growth temperature [49]. Conversely, the warmest treatment (28˚C) was chosen to represent a predicted 2–3˚C increase in mean ocean temperature for the northwest Atlantic shelf [50]. A summary of the duration and the conditions applied during each trial is listed in Table 1.

Treatment seawater was acidified by continuously bubbling mixes of air:100% $CO_2$ into the bottom of each rearing vessel using gas proportioners (ColeParmer®). To maintain low, current-day $pCO_2$ conditions, metabolically produced $CO_2$ was scrubbed from treatment seawater by injecting $CO_2$-stripped air into diffuser tubing at the rearing tank bottom. $CO_2$ stripping was achieved by forcing compressed air through a series of cylinders containing granular soda lime (AirGas®). Rearing vessels were monitored daily for $pH_{NIST}$ and temperature using a handheld pH electrode with an imbedded temperature thermistor (Hach® Intellical PHC281 pH electrode with HQ11D handheld pH/ORP meter, calibrated bi-weekly using two-point

**Table 1. Summary of four long-term trials rearing *M. menidia*.**

| Trial | Fert. date | Temp | $pCO_2$ levels | Replicate tanks | Days reared | Final N | Final traits |
|---|---|---|---|---|---|---|---|
| 1* | 5/3/2015 | 17˚ | 450, 2200 | 2 | 135 | 229–282 | TL, wW, sex ratio |
| 2 | 5/19/2016 | 17˚ | 450, 2200 | 2 | 135 | 191–234 | TL, wW, sex ratio |
| 3 | 5/3/2016 | 24˚ | 450, 2200 | 2 | 110 | 149–199 | TL, wW, sex ratio |
| 4 | 6/29/2017 | 24˚,28˚ | 450, 2200 | 1 | 88(28˚), 103(24˚) | 121–189 | TL, wW |

Offspring were reared under two $pCO_2$ conditions (μatm) and three temperatures (˚C). Days reared was quantified from fertilization to the final sample.

* Note that trial 1 fish were resampled from Murray et al. 2017.

NIST buffers). Continuous bubbling ensured that dissolved oxygen conditions remained at ~100% saturation. Temperature conditions were maintained by thermostats (Aqualogic®) controlling submersible heaters or in-line chillers (DeltaStar®).

We used pH and total alkalinity ($A_T$) as the two directly measured carbonate parameters to calculate treatment $pCO_2$ levels. At three time points during each rearing trial, 300-ml seawater samples were drawn from each rearing tank and filtered (to 10 μm) into borosilicate bottles. Salinity was measured at the time of collection by a refractometer. Bottles were stored in the dark at 3°C, and within two weeks of sampling duplicate measurements of $A_T$ were made on each seawater sample by endpoint titration (G20 Potentiometric Titrator, Mettler Toledo®). The accuracy (within ±1%) of our titration methodology was calibrated and confirmed by using Dr. Andrew Dickson's certified reference material for $A_T$ in seawater (Batch Nrs. 147, 162, and 164, University of California San Diego, Scripps Institution of Oceanography, https://www.nodc.noaa.gov/ocads/oceans/Dickson_CRM/batches.html). CO2SYS (V2.1, http://cdiac.ornl.gov/ftp/co2sys) was used to calculate the partial pressure and fugacity of $CO_2$ ($pCO_2$, $fCO_2$; μatm) as well as dissolved inorganic carbon ($C_T$; μmol kg$^{1-}$) and carbonate ion concentration ($CO_3^{2-}$; μmol kg$^{1-}$) from measured values of $A_T$, pH, temperature, and salinity using K1 and K2 constants from [51] refitted by [52] and [53] for $KHSO_4$. An overview of pH and carbonate chemistry measurements for each experiment is given in Table 2.

## Field sampling and fertilization

Experimental protocols were approved by the University of Connecticut Institutional Animal Care and Use Committee (Protocol Nr. A17-043), and the investigators received annual trainings for best practices in fish care. No additional permits were required for the collection of wild *M. menidia* or for access to our collection site. Experimental offspring were produced

**Table 2. Measurements of carbon chemistry and temperature from long-term $CO_2$ exposure experiments on *M. menidia*.**

| Trial | Tank | Temp treatment | $pCO_2$ treatment | Temp | pH | $pCO_2$ | Sal | $A_T$ | $C_T$ | $fCO_2$ | $CO_3^{2-}$ |
|---|---|---|---|---|---|---|---|---|---|---|---|
| 1 | 1 | 17 | 450 | 17.3±0.3 | 8.06±0.13 | 500±7 | 31 | 2,112±7 | 1,958±7 | 498±7 | 116.8±1.6 |
| | 2 | 17 | 450 | 17.2±0.6 | 8.07±0.12 | 499±7 | 31 | 2,110±1 | 1,956±1 | 497±7 | 116.6±1.4 |
| | 3 | 17 | 2200 | 17.5±0.4 | 7.42±0.11 | 2,295±65 | 31 | 2,102±10 | 2,138±13 | 2,287±65 | 31.3±0.6 |
| | 4 | 17 | 2200 | 17.5±0.4 | 7.43±0.12 | 2,283±95 | 31 | 2,123±27 | 2,158±24 | 2,275±94 | 32.2±1.8 |
| 2 | 5 | 17 | 450 | 17±0.3 | 8.07±0.07 | 471±4 | 31 | 2013±18 | 1862±17 | 469±5 | 112±1 |
| | 6 | 17 | 450 | 17±0.2 | 8.07±0.07 | 472±6 | 31 | 2007±25 | 1858±23 | 470±6 | 111±2 |
| | 7 | 17 | 2200 | 17.2±0.3 | 7.47±0.08 | 2084±46 | 31 | 2008±44 | 2035±45 | 2077±47 | 32±1 |
| | 8 | 17 | 2200 | 17.2±0.3 | 7.48±0.08 | 2055±31 | 31 | 2009±30 | 2035±30 | 2048±30 | 32±1 |
| 3 | 9 | 24 | 450 | 23.9±1 | 8.1±0.08 | 463±3 | 31 | 2041±21 | 1840±17 | 461±3 | 146±3 |
| | 10 | 24 | 450 | 24±1 | 8.1±0.08 | 462±8 | 31 | 2023±28 | 1822±30 | 460±7 | 143±3 |
| | 11 | 24 | 2200 | 24.2±0.8 | 7.49±0.06 | 2192±25 | 31 | 2058±9 | 2044±32 | 2185±25 | 41±2 |
| | 12 | 24 | 2200 | 24.2±0.8 | 7.5±0.06 | 2113±20 | 31 | 2055±27 | 2053±20 | 2106±20 | 43±1 |
| 4 | 13 | 24 | 450 | 23.7±0.6 | 8.11±0.22 | 460±6 | 30 | 2057±16 | 1861±16 | 458±6 | 144±2 |
| | 14 | 24 | 2200 | 23.7±0.6 | 7.47±0.10 | 2323±40 | 30 | 2065±27 | 2079±28 | 2315±40 | 38±1 |
| | 15 | 28 | 450 | 27.7±0.6 | 8.12±0.17 | 459±13 | 31 | 2104±76 | 1865±59 | 458±13 | 172±15 |
| | 16 | 28 | 2200 | 27.8±0.7 | 7.50±0.18 | 2289±57 | 31 | 2132±83 | 2123±77 | 2282±57 | 49±5 |

Mean (±s.d.) pH (NIST) and temperature (°C) were derived from daily measurements by handheld electrodes. Mean (±s.d.) salinity, total alkalinity ($A_T$; μmol kg$^{-1}$), dissolved inorganic carbon ($C_T$; μmol kg$^{-1}$), partial pressure and fugacity of $CO_2$ ($pCO_2$; $fCO_2$; μatm), and carbonate ion concentration ($CO_3^{2-}$; μmol kg$^{-1}$) were quantified from replicated seawater samples. Salinity was measured via refractometer, $A_T$ from endpoint titrations, and $pCO_2$, $C_T$, $fCO_2$ and $CO_3^{2-}$ were calculated in CO2SYS.

from four collections of wild, spawning ripe Atlantic silversides during their spring reproductive seasons in 2015, 2016, and 2017 (S1 Table). All spawners were collected by beach seine (30 × 2 m) from Mumford Cove, CT (41˚ 19.25' N, 72˚ 1.09'W), a shallow embayment that opens to eastern Long Island Sound. Spawning ripe adults were transported to the Rankin Seawater Facility (University of Connecticut Avery Point) where they were separated by sex (by applying light abdominal pressure and inspecting the initial flow of gametes) and held for 24–48 h at low densities (<20 fish) in large aerated tanks (50 L, 17˚-20˚C, ambient $p$CO$_2$, no food). For each of the four fertilizations, embryos were produced by strip-spawning according to established protocols for this species (S1 Table) [28, 54, 55]. Briefly, eggs from all females were stripped together into shallow plastic trays lined with 1-mm carbon fiber window screening. Milt from all males was collected into a single 300-ml plastic cup, mixed, and then poured over eggs. Fertilized eggs, attached to window screening via chorionic filaments, were then disinfected for 15 min in a 100-ppm buffered povidone-iodine solution (Ovadine, Western Chemical, Inc®) before distribution to rearing tanks. Spawned adults were euthanized with an overdose of MS-222 and the number and mean length of spawners used per sex are provided in S1 Table.

## Experimental rearing

Experimental rearing methods closely followed protocols detailed in Murray et al. (2017). Trials 1–3 were conducted in four 700-L main tanks (N = 2 per CO$_2$ treatment). For trial 4, space restrictions allowed only one rearing tank per CO$_2$ × temperature treatment (Table 2). Within 2 hrs of fertilization, >600 fertilized embryos were randomly distributed into 3–4 20-L circular rearing vessels situated inside the 700-L main rearing tanks. At this stage, main tanks were filled with 300-L of filtered (to 1μm) and UV-sterilized seawater from the Long Island Sound (salinity ~31 psu). Treatment seawater was continuously filtered for solid and nutrient waste by 4-stage canister biofilters and 9-watt UV sterilizers (Polar Aurora®), then pumped directly into individual rearing vessels, which were outfitted with flow-through screening. Rearing vessels were tested daily for levels of nitrogenous waste (Saltwater Master Test Kit, API®) to maintain ammonia concentrations at uncritical levels below 0.25 ppm. All experiments were conducted at light conditions of 15h L:9h D. Rearing tanks were monitored daily for indicators of fish stress in response to experimental tank conditions (e.g., heavy and irregular breathing, erratic swimming behavior, loss of orientation, disease). If any of these signs appeared, all water parameters were immediately checked, and if the individual fish failed to recover within 24 h, they were removed from the rearing container and euthanized with an overdose of MS-222 (Western Chemical, Inc.).

Upon hatching, larvae were immediately provided *ad libitum* rations of newly hatched brine shrimp nauplii (*Artemia salina*, San Francisco strain, brineshrimpdirect.com) and small rations of a powdered weaning diet (Otohime Marine Fish Diet, size A1, Reed Mariculture®) to stimulate feeding. Thereafter, larvae were provided *ad libitum* daily rations of newly hatched nauplii only. Rearing vessels were cleaned daily for solid waste. When larvae reached ~10 mm total length (TL) they were counted and distributed at equal densities into three 50-L rearing tubs per main tank (200–250 larvae per tub). During trials 1–3, tubs were also sub-sampled for TL measurements (N ≥ 16), and larvae were immediately euthanized with an overdose of MS-222 and preserved in a 10% formaldehyde/freshwater solution saturated with sodium tetraborate buffer. TL was measured (nearest 0.01 mm) via calibrated microscope images using Image Pro Premier (V9.0, Media Cybernetics®). Rations of newly hatched nauplii were standardized to the known number of juveniles per tub. Larval feed was supplemented with small rations of powered food (Otohime Marine Fish Diet, size B1, Reed Mariculture®) in preparation for a

diet shift. Tubs were checked daily for mortalities, which were counted and discarded, siphoned for waste, and 10% of the treatment seawater was exchanged. Larval mortality rates were typical for this species and similar across treatments [28, 30, 48].

After ~1200 degree-days of rearing (degree-day = rearing temperature $^*$ days reared post-hatch, ddph), surviving juveniles were counted, and sub-samples euthanized with an overdose of MS-222 and preserved for TL measurements via calipers (N $\geq$ 10, nearest 0.1 mm). The remaining fish were placed back into their original main tanks containing 350 L of seawater. Equal starting densities of juveniles were maintained *within* each trial, but *across* trials densities varied from 154–626 fish per tank. Daily rations of powdered diet (Otohime Marine Fish Diet, size B1-B2, Reed Mariculture®) were standardized to 20% of the estimated daily dry weight (dW) biomass per tank. Dry weight biomass was estimated from the known number of fish per tank, mean TL based on sub-samples, and a known TL:dW relationship for *M. mendia* [28]. Ration levels were then increased daily at the same rate within trials based on previously published long-term growth data for this species [56]. Subsequent subsamples for TL measurements were taken over time to recalibrate ration levels (S2 Table). Powdered food was continuously supplied during daylight hours via belt feeders. Tanks were siphoned for waste and 10% of the treatment seawater was exchanged daily.

Rearing trials were terminated depending on the temperature treatment after 2,074–2,496 ddph (83–122 dph, Table 1), which is approximately a third of the lifespan of *M. menidia* [28]. Surviving fish within a trial were euthanized on the same day and measured for TL (nearest 0.1 mm) and wet weight (wW, nearest 0.01 g). For trial 1, half of the fish per rearing tank were randomly sampled for this analysis. For trials 2 and 3, all but 50 randomly selected fish per tank were sampled for measurements. All fish reared during trial 4 were sampled when the experiment was terminated. The sex of juveniles reared at 17˚C (trials 1, 2) and 24˚C (trial 3) was determined by visual inspection of gonads with a dissecting microscope (8× magnification) and confirmed if necessary, by examining gonadal tissue for developing oocytes with a compound microscope (200× mag). The researcher who sexed the fish was blind to the treatment conditions. See Table 3 for final sample sizes.

### Response traits and statistical analyses

Juvenile survival was quantified for each rearing tank from ~1200 ddph to experiment termination. Percent survival was logit transformed (the natural log of percent/(1-percent)) and we tested for significant effects of $p$CO$_2$ level within trial 1–3 using independent samples t-test [57]. Individuals subsampled during the course of the experiment were measured only for TL (0.1 mm), but juveniles at the end each trial were measured for TL and wW, from which we calculated Fulton's condition factor ($k$):

$$k = 100 \times \text{wW}_{(g)} \times \text{TL}^{-3}_{(cm)}$$

A Pearson's chi-squared test was used to compare the percent of female fish between ambient vs. high CO$_2$ treatments for each trial. For trials 1–3, linear mixed-effects models (LMM) were constructed to test for sex-specific $p$CO$_2$ effects on growth (TL, wW, and $k$). To account for a common rearing environment, tank was included as a random effect:

$$\text{TL}\,(\text{wW},\,k) = p\text{CO}_2 + \text{sex} + p\text{CO}_2 \times \text{sex} + \text{tank} + \text{error}.$$

We also analyzed how trait frequency distributions varied between $p$CO$_2$ treatments by implementing a series of shift functions [58]. Within each trial (1–4), measurements of TL, wW, and $k$ were pooled from replicate tanks and five quantiles (0.1, 0.25, 0.5, 0.75, and 0.9) from each treatment were computed using a Harrel-Davis quantile estimator [59]. For each

**Table 3. Summary data for juvenile *M. menidia* from long-term CO$_2$ exposure experiments.**

| Trial | Temp (˚C) | Final age | $p$CO$_2$ (µatm) | Tank | Sex | N | TL (mm) | wW (mg) | Fulton's $k$ |
|---|---|---|---|---|---|---|---|---|---|
| 1 | 17˚ | 135 | 450 | 1 | F | 124 | 42.2±6.0 | 318±117 | 0.41±0.05 |
| | | | | | M | 133 | 39.6±5.4 | 263±94 | 0.41±0.05 |
| | | | | 2 | F | 98 | 42.4±5.6 | 309±112 | 0.39±0.04 |
| | | | | | M | 130 | 42.3±5.8 | 306±116 | 0.39±0.03 |
| | | | 2,200 | 3 | F | 120 | 42±5.5 | 321±111 | 0.42±0.05 |
| | | | | | M | 162 | 37.3±6.1 | 236±107 | 0.43±0.12 |
| | | | | 4 | F | 107 | 41.1±5.3 | 320±112 | 0.45±0.06 |
| | | | | | M | 158 | 38.8±5.7 | 274±113 | 0.45±0.05 |
| 2 | 17˚ | 135 | 450 | 5 | F | 101 | 50.8±5.2 | 613±189 | 0.45±0.04 |
| | | | | | M | 133 | 48.6±5.5 | 540±178 | 0.45±0.03 |
| | | | | 6 | F | 111 | 48.6±4.5 | 542±176 | 0.46±0.03 |
| | | | | | M | 113 | 47.2±5.3 | 499±172 | 0.46±0.03 |
| | | | 2,200 | 7 | F | 104 | 44.0±5.2 | 438±160 | 0.49±0.04 |
| | | | | | M | 113 | 42.5±4.7 | 389±129 | 0.49±0.04 |
| | | | | 8 | F | 97 | 46.4±4.8 | 505±162 | 0.49±0.04 |
| | | | | | M | 94 | 45.2±4.6 | 472±144 | 0.50±0.04 |
| 3 | 24˚ | 110 | 450 | 9 | F | 19 | 54.5±7.0 | 994±409 | 0.58±0.04 |
| | | | | | M | 180 | 55.1±6.1 | 1012 ±329 | 0.58±0.04 |
| | | | | 10 | F | 22 | 56.8±5.1 | 1080±296 | 0.57±0.03 |
| | | | | | M | 170 | 56.2±6.9 | 1082±387 | 0.58±0.04 |
| | | | 2,200 | 11 | F | 15 | 53.7±3.8 | 893±177 | 0.57±0.03 |
| | | | | | M | 134 | 53.4±5.6 | 899±274 | 0.57±0.04 |
| | | | | 12 | F | 19 | 57.5±7.0 | 1195±386 | 0.60±0.04 |
| | | | | | M | 158 | 55.4±4.7 | 1023±276 | 0.59±0.04 |
| 4 | 24˚ | 103 | 450 | 13 | - | 189 | 58.2±5.5 | 1269±339 | 0.62±0.05 |
| | | | 2,200 | 14 | - | 161 | 57.9±4.7 | 1230±295 | 0.62±0.03 |
| | 28˚ | 88 | 450 | 15 | - | 121 | 48.5±4.6 | 776±202 | 0.67±0.05 |
| | | | 2,200 | 16 | - | 128 | 47±4.5 | 714±202 | 0.67±0.04 |

Data are displayed as rearing tank means (±s.d.). Final age was quantified as the number of days from fertilization to final sample.

trait, quantile estimates from the low $p$CO$_2$ treatment were subtracted from the high $p$CO$_2$ distribution, and 95% confidence intervals (CIs) for quantile differences were calculated using a bootstrap (N = 1,000) estimation of the standard error of the quantile [60]. Significant CO$_2$ effects on quantile differences were assumed if bootstrapped 95% CIs did not include zero. Significance levels for the 5 quantile comparisons were adjusted for multiple comparisons within a single test via Hochberg's method [61].

To evaluate time-dependent effects of high $p$CO$_2$ exposure, we employed LMMs to test for CO$_2$ effects on the TL of each group of sub-sampled offspring (S2 Table) using the model:

$$TL = p\text{CO}_2 + \text{tank} + \text{error}.$$

All statistical analyses were performed in R (version 3.5.3) using RStudio (version 1.2.1). LMMs were run using the *lme4* [62] package using maximum likelihood estimates for fixed effects. Significance levels were determined by Satterthwaite's method via the *lmertest* package [63]. The normality and variance homogeneity of model residuals were assessed by visual inspection of QQ plots and residual boxplots, respectively [64]. The shift analysis and plots were generated using the R package *rogme* e [65]. We used Cohen's d to calculate CO$_2$ effect

sizes (±95% CIs) using the R package *effsize* [66] where negative values indicate a trait reduction under high $pCO_2$ [67].

## Results

### Trials 1–3 $pCO_2$ effects on sex ratio

A summary of sex ratio and body size data of juveniles is listed in Table 3. During trial 1, female sex ratios at 17°C were not significantly different between juveniles reared at 450 μatm (46±4%) and 2,200 μatm $pCO_2$ (41±2%). A similar result was observed after trial 2 (Fig 1),

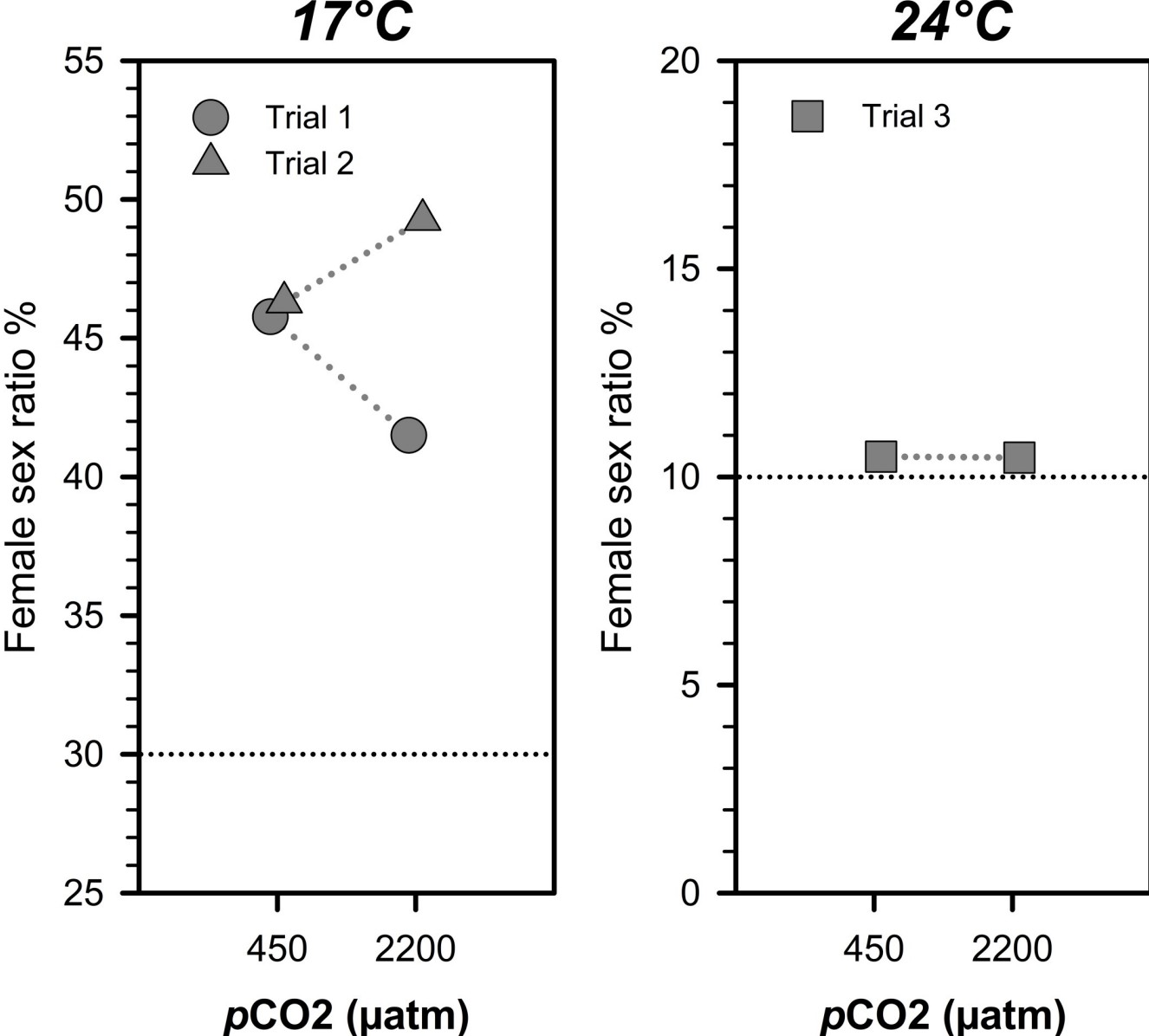

**Fig 1. Female sex ratios from trials 1–3.** The mean female sex ratio (F/(F + M)) of juvenile *M. menidia* reared under 450 and 2,200 μatm $pCO_2$ at 17° and 24°C. Dotted lines connect treatment means within trials. Horizontal black lines indicate the temperature dependent female sex ratios predicted for the experimental source populations by Conover & Heins (1987).

where the proportion of females was roughly equal between $p$CO$_2$ treatments (450 µatm: 46 ±5%; 2,200 µatm: 49±2%). At 24˚C, the proportion of females was similarly low at ambient (450 µatm: 11±1%) versus high $p$CO$_2$ conditions (2,200 µatm:10±1%, Fig 1).

## Long-term $p$CO$_2$ × sex effects on growth

**Trial 1.** Juvenile survival (mean ± s.d.) was similar in ambient (84±2%) and high $p$CO$_2$ (88±1%) treatments. The TL of juveniles from high $p$CO$_2$ was significantly lower compared to ambient conspecifics (LMM, $p$ = 0.034, Table 4, Fig 2A). Female fish were significantly longer than males (Tables 3 and 4), and the LMM detected a significant $p$CO$_2$ × sex interaction ($p$ = 0.002, Table 4), indicating that male TL was more negatively impacted by high $p$CO$_2$ exposure than female TL (Table 5). Shift analysis revealed a uniform and significant reduction in TL under high $p$CO$_2$ across the entire TL distribution (Fig 2A). Juvenile wW was also significantly affected by a $p$CO$_2$ × sex interaction (LMM, $p$ = 0.009, Table 4), but the male-specific high $p$CO$_2$ effect size was small (>-0.30, Table 5). Female fish were significantly heavier than males (Tables 3 and 4). Shift analysis showed that only the lower weight quantiles, largely represented by male fish, were significantly different between $p$CO$_2$ treatments (Fig 2B). In

**Table 4. LMM results for trials 1–3.**

| Trial | Temp (˚C) | Trait | Factor | Num. df | Den. df | F | $p$ |
|---|---|---|---|---|---|---|---|
| 1 | 17 | | $p$CO$_2$ | 1 | 3.99 | 13.987 | **0.034** |
| | | TL | Sex | 1 | 1029.42 | 0.035 | **<0.001** |
| | | | $p$CO$_2$ × sex | 1 | 1029.42 | 0.825 | **0.002** |
| | | | $p$CO$_2$ | 1 | 4.26 | 2.157 | 0.330 |
| | | wW | Sex | 1 | 1028.32 | 0.080 | **<0.001** |
| | | | $p$CO$_2$ × sex | 1 | 1028.32 | 0.866 | **0.009** |
| | | | $p$CO$_2$ | 1 | 4.04 | 10.387 | **0.019** |
| | | $k$ | Sex | 1 | 1029.91 | <0.001 | 0.301 |
| | | | $p$CO$_2$ × sex | 1 | 1029.91 | 0.009 | 0.167 |
| 2 | 17 | | $p$CO$_2$ | 1 | 3.99 | 15.519 | **0.017** |
| | | TL | Sex | 1 | 862.37 | 20.992 | **<0.001** |
| | | | $p$CO$_2$ × sex | 1 | 862.37 | 0.480 | 0.488 |
| | | | $p$CO$_2$ | 1 | 3.98 | 8.812 | **0.041** |
| | | wW | Sex | 1 | 862.44 | 19.582 | **<0.001** |
| | | | $p$CO$_2$ × sex | 1 | 862.44 | 0.521 | 0.474 |
| | | | $p$CO$_2$ | 1 | 3.89 | 226.652 | **<0.001** |
| | | $k$ | Sex | 1 | 861.79 | 03069 | 0.792 |
| | | | $p$CO$_2$ × sex | 1 | 861.79 | 0.012 | 0.913 |
| 3 | 24 | | $p$CO$_2$ | 1 | 8.06 | 0.287 | 0.607 |
| | | TL | Sex | 1 | 712.09 | 1.061 | 0.303 |
| | | | $p$CO$_2$ × sex | 1 | 712.09 | 0.836 | 0.361 |
| | | | $p$CO$_2$ | 1 | 6.75 | 0.268 | 0.621 |
| | | wW | sex | 1 | 712.05 | 1.131 | 0.288 |
| | | | $p$CO$_2$ × sex | 1 | 712.05 | 1.650 | 0.199 |
| | | | $p$CO$_2$ | 1 | 6.38 | 0.139 | 0.722 |
| | | $k$ | sex | 1 | 712.00 | <0.001 | 0.989 |
| | | | $p$CO$_2$ × sex | 1 | 712.00 | 0.268 | 0.605 |

Summary statistics for LMM testing $p$CO$_2$ and sex effects (fixed) on the final TL, wW, and Fulton's $k$ of *M. menidia* juveniles reared during Trials 1–3. Numerator (num.) and denominator (den.) degrees of freedom are shown and significant $p$ values are denoted in bold.

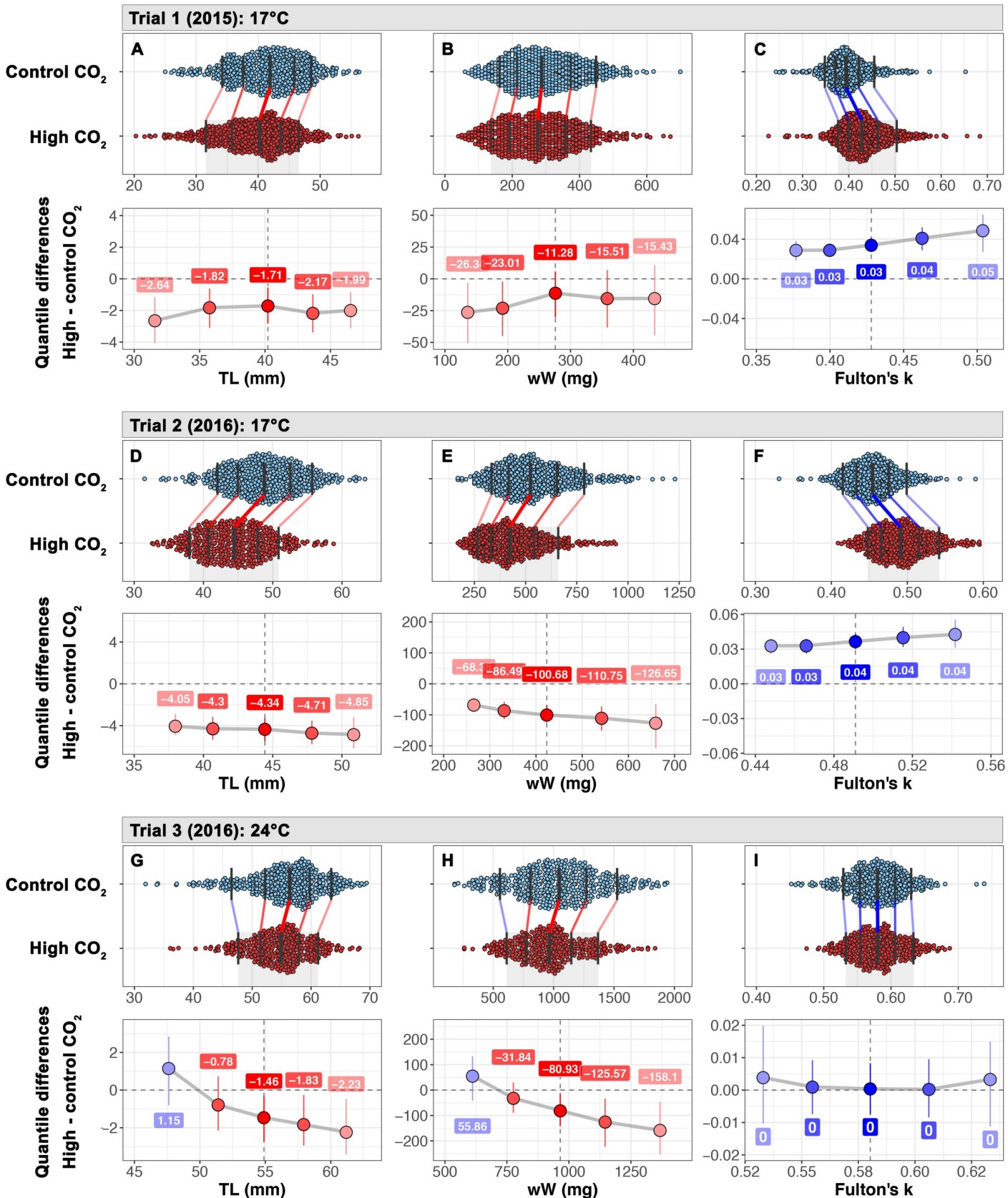

**Fig 2. Shift functions and quantile differentials for trials 1–3.** *M. menidia*. Shift functions for trials 1 (A-C), 2 (D-F), and 3 (G-I) are denoted by different letters. Upper panels show frequency density distributions as colored dots (blue: 450 μatm; red: 2,200 μatm). Black vertical bars overlaying each distribution indicate the .1, .25, .5, .75, and .9 quantiles. Quantile shifts are indicated by connecting lines where red lines indicate a reduction in trait value and blues denote a positive shift. The lower panels show quantile differentials (high $p\mathrm{CO_2}$ –ambient $p\mathrm{CO_2}$) and bootstrapped 95% CIs. Dots are color coded to indicate a negative (red) or positive effect of high $p\mathrm{CO_2}$ on the trait value. The size of the quantile shift is denoted in color boxes above or below the colored dots.

contrast to body size, juveniles from 2,200 μatm $p\mathrm{CO_2}$ exhibited significantly higher Fulton's $k$ values compared to ambient fish (LMM, $p$ = 0.019, Tables 3 and 4). This effect did not vary by sex (Tables 3–5) and was uniform across the frequency distribution (Fig 2C)

**Trial 2.** Juvenile survival at 17°C was similarly high under ambient (98±1%) and high $p\mathrm{CO_2}$ (96±2%). Again, exposure to high $p\mathrm{CO_2}$ conditions significantly reduced TL (LMM, $p$ = 0.017, Table 4) and wW (LMM, $p$ = 0.041, Table 4). While female fish were significantly longer and heavier (Tables 3 and 4), the effect of high $p\mathrm{CO_2}$ on growth was not sex-dependent this time (Table 4). When averaged between sexes, the negative $p\mathrm{CO_2}$ effect size on TL and wW more than doubled from trial 1 to trial 2 (TL: -0.83, wW: -0.58, Table 5). The shift analysis showed that quantile differences for TL and wW were significant across frequency distributions (Fig 2D and 2E). Consistent with trial 1, Fulton's $k$ was again significantly higher in juveniles reared under high $p\mathrm{CO_2}$ (LMM, $p$ < 0.001, Table 4), the effect was independent of sex (Tables 3 and 4) and statistically uniform across the frequency distribution (Fig 2F).

**Trial 3.** Juvenile survival at 24°C was not affected by $p\mathrm{CO_2}$ level (ambient: 96±3%; high $p\mathrm{CO_2}$: 92±8%). In contrast to the negative effects observed at 17°C, juvenile TL, wW, and $k$ were all statistically unaffected by $p\mathrm{CO_2}$ level and sex (Table 4). However, the shift analysis indicated that high $p\mathrm{CO_2}$ effects were not uniform across TL and wW frequency distributions. While the lower size quantiles were unaffected by $p\mathrm{CO_2}$ level, the 0.5, 0.75, and 0.9 quantiles

**Table 5. Sex-specific high $p\mathrm{CO_2}$ effect sizes.**

| Trial | Trait | Sex | Cohen's d |
|---|---|---|---|
| 1 | TL* | Female | -0.11±0.19 |
| | | Male | -0.50±0.17 |
| | wW | Female | 0.06±0.19 |
| | | Male | -0.27±0.16 |
| | $k$ | Female | 0.59±0.19 |
| | | Male | 0.58±0.17 |
| 2 | TL | Female | -0.88±0.20 |
| | | Male | -0.82±0.19 |
| | wW | Female | -0.60±0.20 |
| | | Male | -0.59±0.19 |
| | $k$ | Female | 1.02±0.21 |
| | | Male | 1.10±0.20 |
| 3 | TL | Female | 0.03±0.46 |
| | | Male | -0.19±0.15 |
| | wW | Female | 0.06±0.46 |
| | | Male | -0.24±0.16 |
| | $k$ | Female | 0.17±0.46 |
| | | Male | 0.03±0.15 |

Effect sizes were quantified using Cohen's d (treatment means ± 95% CI). Negative values indicate a trait reduction under in juveniles from high $p\mathrm{CO_2}$ conditions relative to ambient conspecifics.

*Indicates a significant difference in effect size between sexes.

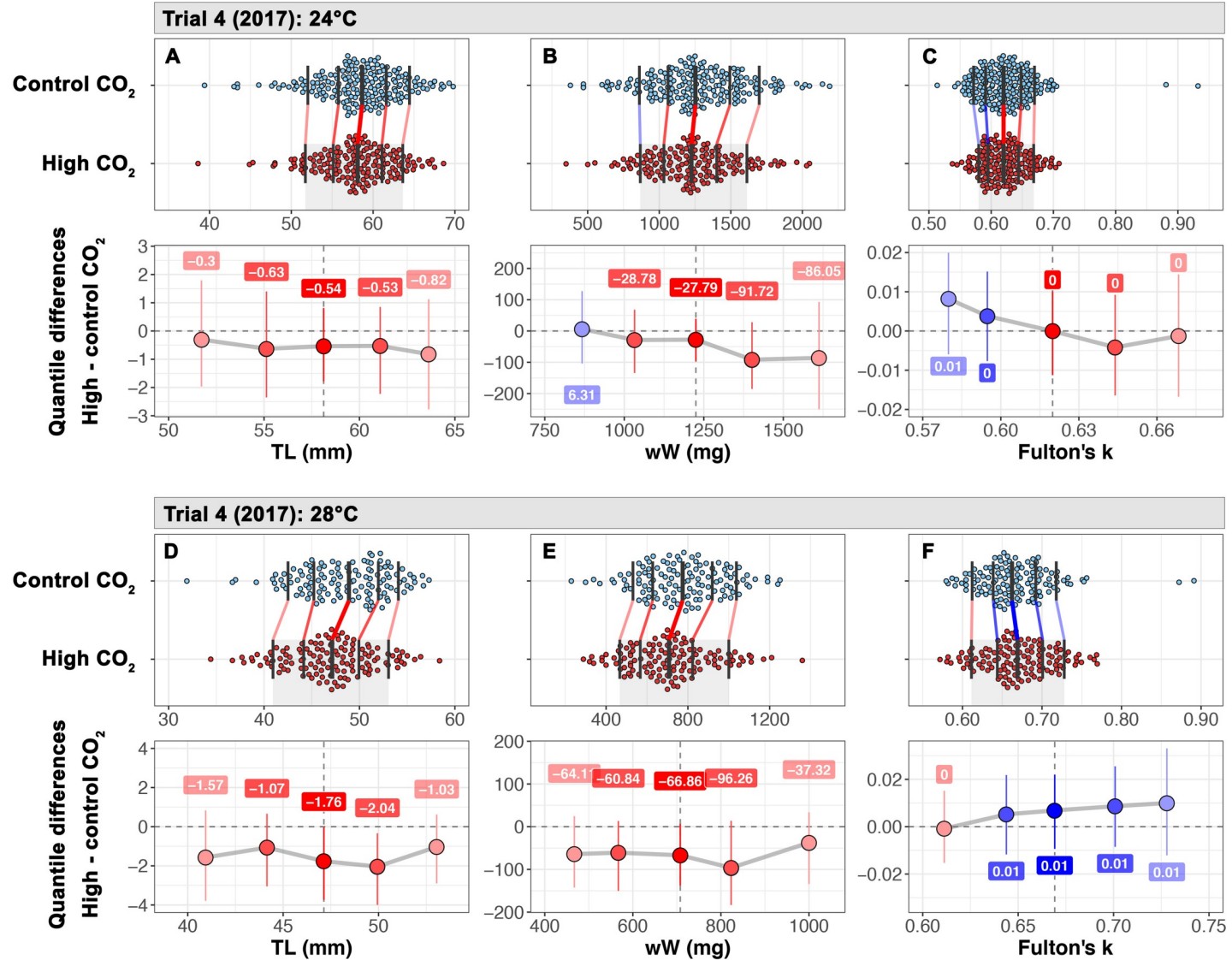

**Fig 3. Shift functions and quantile differentials for trial 4.** *M. menidia*. Temperature treatments are indicated by differing letters (24°C: A-C; 28°C: D-F). Upper panels show frequency density distributions as colored dots (blue: 450 μatm; red: 2,200 μatm). Black vertical bars overlaying each distribution indicate the .1, .25, .5, .75, and .9 quantiles. Quantile shifts are indicated by connecting lines where red lines indicate a reduction in trait value and blues denote a positive shift. The lower panels show quantile differentials (high $p$CO$_2$ –ambient $p$CO$_2$) and bootstrapped 95% CIs. Dots are color coded to indicate a negative (red) or positive effect of high $p$CO$_2$ on the trait value. The size of the quantile shift is denoted in color boxes above or below the colored dots.

shifted lower in the high compared to ambient $p$CO$_2$ distribution (Fig 2G and 2H). By contrast, the effect $p$CO$_2$ on Fulton's $k$ was neutral across the frequency distribution (Fig 2I).

**Trial 4.** Juvenile survival was high across rearing tanks (95–99%). At 24°C, TL and wW distributions were shifted to lower sizes and weights compared to the ambient $p$CO$_2$ treatment but the effect was not significant across the distribution (Fig 3A–3C). There was no CO$_2$ effect on Fulton's $k$. However, for juveniles reared at 28°C long-term exposure to 2,200 μatm $p$CO$_2$ resulted in an average reduction in TL and wW compared to ambient $p$CO$_2$ juveniles and the effect was significant at the median and .75 quantiles (Fig 3D and 3E). The overall high $p$CO$_2$ effect size was small (>-0.40, Table 5). Fulton's k was unaffected by $p$CO$_2$ level at 28°C (Fig 3F).

## Trials 1–4 $pCO_2$ × age effects

S2 Table contains summary data for sub-sampled offspring. At 17°C, we found that the negative effect size of high $pCO_2$ on TL increased with age (Cohen's d, 16–21 dph: -0.32, 68–69 dph: -0.62, 100–103 dph: -0.80), but this $CO_2$ effect was only significant after more than 100 days of continuous exposure to acidified conditions (LMM, trial 1: $p = 0.021$, trial 2: $p < 0.001$, Fig 4A). By contrast, at 24°C and 28°C there were no $CO_2$ effects on TL of sub-sampled offspring over time (Fig 4B and 4C).

## Discussion

Potential sex-specific responses of organisms to high $pCO_2$ environments remain an understudied aspect of ocean acidification research [37]. Since fish display a range of sexual variation in physiology, behavior, and bioenergetics [68] that are also impacted by elevated $pCO_2$ [6, 8, 14], sex may influence how individual fish respond to OA conditions. Here, we examined sex-specific growth in Atlantic silverside juveniles reared at 17° and 24°C, and our findings not did support the hypothesis of higher female than male $CO_2$ sensitivity. Actually, males in trial 1 were disproportionally impacted by high $pCO_2$ at 17°C, but his effect was not reproduced in subsequent trials. Furthermore, we did not find evidence that juvenile sex ratios differed between $pCO_2$ treatments, hence, seawater $pCO_2$/pH conditions are unlikely to impact environmental sex determination in silverside larvae. The female sex ratios were consistent with previously reported values of ~10% at 24°C and ~45% at 17°C [69].

However, because our findings are limited to pre-spawning individuals, key unknowns regarding sex-specific $CO_2$ effects in mature fish remain. A distinct bioenergetic difference between the sexes concerns the maturation of gametes, given that egg production is generally more costly than sperm [68]. While the juveniles in our study had clearly differentiated gonads, females had yet to begin the more energetically intensive stages of vitellogenesis [70]. Furthermore, sexual dimorphism in size was apparent in this study and is prominent in wild silverside populations [71, 72] as selection for large body size confers a greater reproductive advantage to female fish [38]. As an annual species, juvenile growth in silversides is a key determinant of a female's reproductive output during their only spawning season [38]. Therefore, while growth reductions under high $pCO_2$ were similar or slightly greater in male fish in this study, the reproductive impacts of a smaller body size might be more consequential for female fish. Furthermore, other biochemical or behavioral consequences associated with long-term $CO_2$ acclimation might influence the reproductive output of both sexes [37]. To date, very few studies have quantified $CO_2$ impacts on fish reproductive output and offspring viability, reporting inconsistent outcomes [73, 74]. Further examinations of sex-specific $CO_2$ responses are critically needed, especially if $CO_2$ sensitivity is confounded by the many reproductive strategies employed by fish [70].

Juvenile *M. menidia* reared at 17°C exhibited small but consistent reductions in size under high $pCO_2$ during two experimental years. During trial 3, the linear mixed-effects model did not detect an overall effect of high $pCO_2$ on growth at 24°C, but the shift analyses showed that impacts varied across the TL and wW frequency distributions. While fish from the smallest quantiles were similarly sized, juveniles making up the median, 0.75 and 0.9 quantiles of the high $pCO_2$ distribution were significantly smaller than the same quantiles from ambient $pCO_2$. In fact, these reductions were similar in magnitude to what we observed at 17°C. This suggests that long-term exposure to high $pCO_2$ may still limit growth at optimal thermal conditions by restricting the development of the fastest growing individuals. However, during trial 4 we did not observe the same patten at 24°C, despite that fact that high $pCO_2$ quantile differentials were consistently shifted downward to a smaller size. We also reared offspring at 28°C which

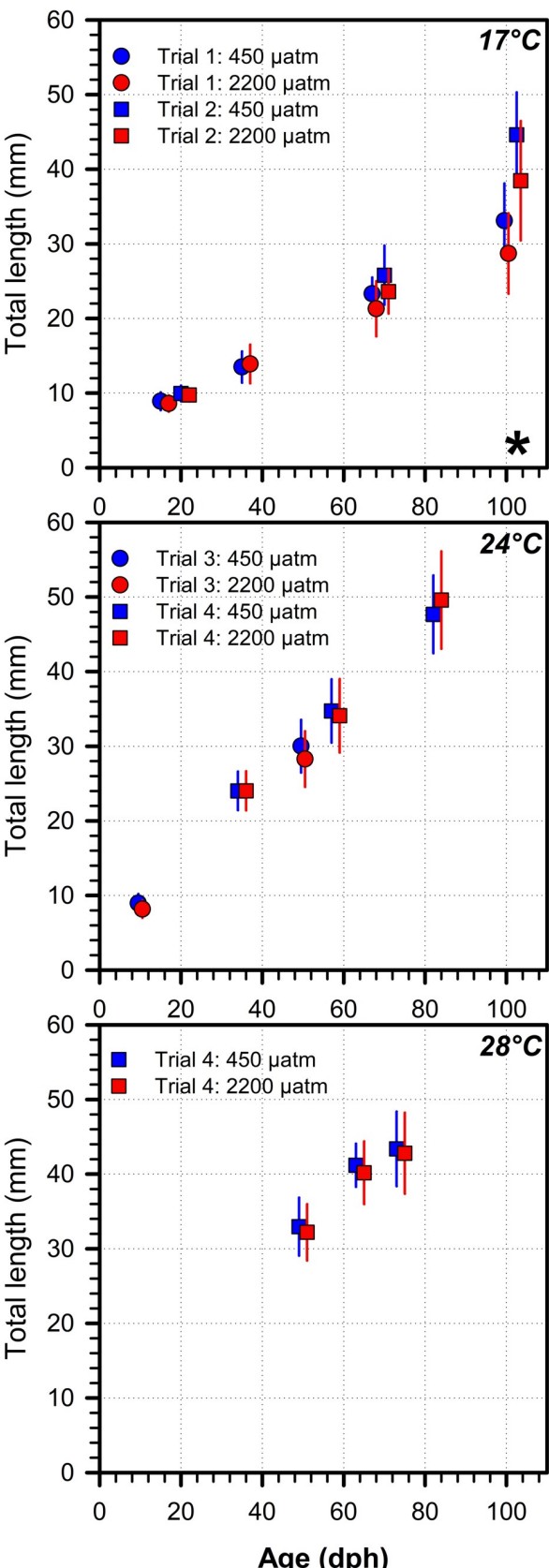

**Fig 4. TL of subsampled juveniles.** *M. menidia*. Mean TL (± s.d.) of all subsampled juveniles reared under two $pCO_2$ conditions (blue: 450 μatm; red: 2,200 μatm) and three temperatures. Significant differences between $pCO_2$ treatment within sampled age groups are denoted by black stars (LMM, $p < 0.05$).

is near the upper thermal limit for positive growth in silversides. The shift analysis showed that quantile reductions under high $pCO_2$ were twice as large than what was observed at 24°C during trial 4, and the reductions were significant for several quantiles. Thus, across trials and temperature treatments, the average juvenile fish from high $pCO_2$ conditions was shorter (-2 to -9%) and weighed less (-3 to -18%) than ambient conspecifics. Interestingly, the percent reductions in whole-animal size observed here are proportionally similar to the increased energetic demands of intestinal tissues isolated from Gulf toadfish (*Opsanus beta*), which showed an 8% increase in energetic consumption and a 13% increase in intestinal bicarbonate secretion when exposed to 1,900 μatm $pCO_2$ [26]. Hence, the reductions in body size observed in this study likely reflect the increased long-term homeostatic costs of life under high $pCO_2$.

Our findings suggest that negative growth responses to high $pCO_2$ show a parabolic relationship with temperature and become stronger at sub-optimal thermal conditions [35]. However, low replication at 28°C limited the power of our analysis, and more data are needed to sufficiently analyze $CO_2$ effects at this upper thermal limit. A similar pattern between temperature and $CO_2$ sensitivity was found in juvenile Atlantic halibut (*Hippoglossus hippoglossus*), where negative growth effects of high $CO_2$ only manifested at the coldest rearing condition [75]. While 17°C is well within the thermal tolerance limits of Atlantic silversides, it is near the lower limit for early life stages to maintain positive growth [76, 77]. Chronic exposure to a low-growth thermal regime that depresses the performance of circulatory and respiratory systems could also compromise the homeostatic mechanisms that buffer against environmental acidosis. These mechanisms require further study as a definitive link between growth, aerobic scope, $CO_2$ and temperature sensitivity has not been established [36, 75].

Despite their reduced length and weight, we found that juveniles reared at 17°C under acidified conditions consistently exhibited higher Fulton's $k$ values than ambient conspecifics. Long-term exposure to high $pCO_2$ conditions caused a greater reduction in average length than weight, hence an increase in Fulton's $k$. While the basis of this increased condition is unknown, it does suggest that acidified environments change the way in which silversides partition resources. Exposure to high $pCO_2$ could also change the shape of developing silversides which would confound condition factor comparisons [78]. Atlantic silversides undergo intense size-selective overwintering mortality where large size paired with increased lipid storage is conducive to higher survival [79]. Therefore, a relatively small $CO_2$ induced reduction in the size at onset of the overwintering period could have larger implications for Atlantic silverside population dynamics, as smaller fish incur higher winter mortality and produce fewer viable offspring the following spring [80]. An increase in Fulton's $k$ might offset the risk of winter starvation, but this would entirely depend on individuals acclimated to high $pCO_2$ actively increasing lipid energy stores [81]. In contrast, higher condition values due to changes in shape are not likely to alleviate overwinter mortality. Our understanding of the relationship between high $pCO_2$ exposure and condition factor would benefit from a detailed analysis of energy composition and form factor [82].

Previous work on *M. menidia* early life stages found growth to be largely unaffected by high $pCO_2$ conditions (2,000–6,000 μatm) across the same range of temperatures examined here (17°-28°C) [48]. This study included considerably longer rearing times and older life-stages, finding that $pCO_2$ effects on size increased over time and became statistically detectable after 100 days of continuous exposure or nearly a third of this species lifespan. To date, studies that

evaluated long-term $CO_2$ effects in fish have often utilized longer-lived species where even months of rearing still amount to only a small fraction of their overall lifespan [73, 74, 83–86]. Our results demonstrate that measurable $CO_2$ effects on growth can be detected after a prolonged exposure over multiple life stages. Another important difference between this and previous long-term experiments was our application of a high $pCO_2$ treatment of 2,200 µatm. By contrast, most long-term studies that have reported neutral growth responses have exposed fish to ~1,000 $pCO_2$ [73, 74, 83–86]. While this may highlight the widespread resiliency of fish to predicted end-of-century $pCO_2$ levels [6], such predictions are generalized for the average global ocean [2]. In contrast, coastal marine systems are already prone to periodic acidification near or in excess of 1,000 $pCO_2$ [47, 87, 88] and future anthropogenic impacts will likely intensify the duration and magnitude of these events [89, 90]. As such, experimenters should strive to apply $pCO_2$ treatments that reflect the likely future conditions of the systems where their model organisms live and reproduce.

Most laboratory studies on fish provide rations at excess levels to remove the potential for confounding effects of uneven feeding between treatments, but this practice may mask the energetic costs associated with $CO_2$ acclimation. For example, the clear relationship between higher temperature and increased feeding is due, in part, to compensate for an increased basal metabolic rate of a warmer environment [91]. Yet, a link between $CO_2$ sensitivity and food availability remains unclear. Most short-term studies on larvae and juveniles have found no interaction between ration level and $CO_2$ sensitivity [31, 33] including in *M. menidia* [32], but acidification did exacerbate starvation rates in *Rachycentron canadum* [34]. In this study, to avoid a potential masking effect of excess food consumption, we provided non-excess rations to post-larval fish (>20 mm) that were standardized to the estimated total daily biomass per rearing tank. Food availability can vary seasonally and across ontogenetic stages such that it plays a critical role determining resiliency to stressors and ultimately how fish populations are structured [11]. Therefore, providing fish with realistic, i.e., non-excess ration levels should be an experimental priority to generate more realistic estimates of long-term $CO_2$ sensitivity.

We found that the $CO_2$ effect on growth at 17°C varied between experimental years. Juveniles reared during the second trial attained a larger final size, and the $CO_2$-induced length reduction doubled from ~2 mm in trial 1 to ~4 mm in trail 2. These differences could have been due to improved rearing methodologies, including improved techniques for the removal of nitrogenous waste and lower fish densities during trial 2. Equally, increased $CO_2$ sensitivity may have arisen from genetic or phenotypic differences between groups of strip-spawned adults [92]. Regardless of the sources of variation, these interannual differences comprise important experimental outcomes. They caution that the complexity of empirical $CO_2$ responses between fish species or populations may reflect methodological differences between laboratories in addition to inherent variations in $CO_2$ sensitivity [22]. Our findings highlight the importance of designing experiments able to detect the cumulative long-term effects of elevated $pCO_2$ on fish bioenergetics. Cooperation amongst research groups to share best practices will maximize the usefulness of inter-laboratory comparisons and produce robust experimental replications [93, 94].

## Supporting information

**S1 Checklist.**
(DOCX)

**S1 Table. Information on adult spawners.** The number and length of Spawning ripe *M. menidia* used to fertilize trials 1–4.
(DOCX)

**S2 Table. Summary statistics of subsampled offspring.** *M. menidia.* Mean (±s.d.) TL and samples sizes (N) of subsampled offspring from trials 1–4.
(DOCX)

**S1 Data.**
(XLSX)

## Acknowledgments

We are grateful to Baumann Lab members J. Snyder, M. Hughes, E. Karamavros, J. Pringle, I. Mayo, C. Dyke, and J. Harrington for assistance in the lab. And to C. Woods for technical assistance. We thank Dr. Guillaume Rousselet for his assistance in developing the shift function plots.

## Author Contributions

**Conceptualization:** Christopher S. Murray, Hannes Baumann.

**Data curation:** Christopher S. Murray.

**Formal analysis:** Christopher S. Murray.

**Funding acquisition:** Hannes Baumann.

**Investigation:** Christopher S. Murray.

**Methodology:** Christopher S. Murray, Hannes Baumann.

**Project administration:** Hannes Baumann.

**Supervision:** Hannes Baumann.

**Validation:** Hannes Baumann.

**Writing – original draft:** Christopher S. Murray.

**Writing – review & editing:** Christopher S. Murray, Hannes Baumann.

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
