## [Decision Letter · Decision Letter 0]

19 Feb 2020

PONE-D-19-33606

Are temperature-specific, long-term growth responses to elevated CO2 sex-specific in fish?

PLOS ONE

Dear Dr. Murray,

Thank you for submitting your manuscript to PLOS ONE. After careful consideration, we feel that it has merit but does not fully meet PLOS ONE’s publication criteria as it currently stands. Therefore, we invite you to submit a revised version of the manuscript that addresses the points raised during the review process.

While the ms contains a massive amount of work, I agree with criticism raised by reviewer 1/2 with respect to pseudoreplication, please clarify. I also agree that it is not possible to run one statistical model over the entire data set, as temp. experiments were conducted in different years. I very much agree with the insightful suggestions of reviewer 1 of how to bring this dataset into a publishable format, so please consider them carefully prior to resubmission.

We would appreciate receiving your revised manuscript by Apr 04 2020 11:59PM. To enhance the reproducibility of your results, we recommend that if applicable you deposit your laboratory protocols in protocols.io, where a protocol can be assigned its own identifier (DOI) such that it can be cited independently in the future. For instructions see: http://journals.plos.org/plosone/s/submission-guidelines#loc-laboratory-protocols

We look forward to receiving your revised manuscript.

Kind regards,

Frank Melzner

Academic Editor

PLOS ONE

Journal Requirements:

2. In your Methods section, please provide additional information regarding the permits you obtained for the work. Please ensure you have included the full name of the authority that approved the collection site access and, if no permits were required, a brief statement explaining why.

3. We noted in your submission details that a portion of your manuscript may have been presented or published elsewhere. [A small portion of the length data analysed for this study has been re-sampled from: Murray CS, Fuiman LA, Baumann H. Consequences of elevated CO2 exposure across multiple life stages in a coastal forage fish. ICES Journal of Marine Science. 2017;74(4):1051-61. doi: 10.1093/icesjms/fsw179.] Please clarify whether this [conference proceeding or publication] was peer-reviewed and formally published. If this work was previously peer-reviewed and published, in the cover letter please provide the reason that this work does not constitute dual publication and should be included in the current manuscript.

Reviewers' comments:

Reviewer's Responses to Questions

**Comments to the Author**

1. Is the manuscript technically sound, and do the data support the conclusions?

Reviewer #1: No

Reviewer #2: Partly

2. Has the statistical analysis been performed appropriately and rigorously? 

Reviewer #1: No

Reviewer #2: No

3. Have the authors made all data underlying the findings in their manuscript fully available?

Reviewer #1: Yes

Reviewer #2: Yes

4. Is the manuscript presented in an intelligible fashion and written in standard English?

Reviewer #1: Yes

Reviewer #2: Yes

5. Review Comments to the Author

Reviewer #1: Review of manuscript PONE-D-19-33606

The manuscript ‘Are temperature-specific, long-term growth responses to elevated CO2 sex-specific in fish?’ proposes to undertake what is unquestionably an area of significant importance within the biological literature, namely the lack of sex-specific responses being considered in a wide range of fields, and in this case specifically the field of climate change. This topic is also incredibly timely with a number of manuscripts starting to address this topic. Unfortunately however, I feel there are a number of significant methodological flaws with this study, which prevent it being possible to recommend for publication in its current form. Below I outline the major concerns, which will hopefully aid the authors in publishing their data, which definitely has value for the ocean acidification field, but certainly not in the way it is currently presented.

Major concerns:

My first major concern is the presentation of this manuscript as testing sex-specific responses across a range of temperature and pCO2 treatments. The title alludes to this fact, and then the introduction is almost entirely focused on the importance and implications of sex disaggregated results. However, in reality the sex-specific responses in this manuscript are restricted to one temperature treatment (17 degrees). The authors argue this is because temperature has a confounding effect on sex ratio at 24 degrees, preventing comparison. However, sex ratio is only presented for one of the two trials in which organisms were exposed to 24 degrees, and completely omitted from the 28 degree treatment. Was it not measured? Equally, whilst organismal numbers are certainly unbalanced at 24 and 28 degrees, the sex-specific response is still surely valid? As in nature where elevated temperatures and elevated pCO2s co-exist, this same male biased sex ratios would surely be present, and thus the possibility of subsequent sex-specific responses even more important for the future viability of this species, especially if any female enhanced sensitivity is noted. Also clearly this male bias with sex ratio is a major concern under climate change scenarios for this species, a point entirely overlooked in this paper. The alignment of this paper as testing sex-specific responses is not founded, and in reality can only be presented IF the 17 degree data is presented alone, hence removing the multi-stressor angle of the paper, due to the omission of meaningful sex disaggregated data from 2 of the 3 temperature treatments.

The second major concern is the authors propose to undertake a long-term multi factorial experiment over three years. Whilst this statement is true, it implies a well-designed, replicated fully factorial multi stressor experiment. In reality each trial, of which there were 4, tested the impact of pCO2 on fish growth under a single temperature, in different years. This approach is not valid. The experimental results obtained for the impact of elevated pCO2 on fish growth at 28 degrees, collected in 2017, are not directly comparable to the data collected in 2015 at 17 degrees. There are clearly confounding impacts that invalidate direct statistical comparison. Equally, the authors themselves claim variability in data collected in subsequent years may be due to an optimisation of experimental methodology during subsequent years. So they themselves appreciate that it is impossible to replicate the experiment in subsequent years identically, so designing the experiment to have each treatment tested in different years is a major concern. Data can certainly be compared within years/treatments, but not across treatments as presented.

The third major concern is the level of replication applied is very difficult to interpret. As presented is appears that embryos are divided in 3-4 circular vessels in a single main rearing tank per treatment. Then this is adjusted to 3x 50l vessels when larvae reach 10mm, distributed in the rearing vessel, before finally animals are released into the rearing vessel at 1200 degree days. This is 70 days, 50 days and 42 days post hatch respectively for 17, 24 and 28 degrees. So based on an experimental duration of 135 days, 107 days, and 88 days for these temperature treatments respectively, over half the experiment is conducted with all larvae in a single experimental rearing tank per trial. The rearing tank is the level at which the treatment is applied, so this is pseudo-replication, and in the case of the 28 degree treatment there is only 1 replicate. Either this is inadequate study design, or it is inadequate explanation of the exact experimental protocol in the methods. Hopefully this is the latter, and further detail can reassure the editor/reviewers/readers that the level of replication and statistical design was appropriate. Simply including tank, or year, as random effects in a LMM in not sufficient justification, when each year contains a single treatment (thus year would be exactly analogous to treatment effect).

My final concern is the water chemistry testing. Water used for the experiment came from Long Island sound, and was distributed in the rearing tanks at 300L. Each day 10% of water was changed from each rearing tank. Meaning essentially 100% water change every 10 days (although in reality not quite the case due to dilution). However, during each trial AT and salinity were measured 3 times, at 17 degrees this is 3 times in 135 days. So in the 45 days between AT measurements, water could have change completely 4 times. Given the known dynamics of carbonate chemistry in nearshore coastal regions, parameterising alkalinity this infrequently is not enough to fully describe the carbonate chemistry in this experiment, even if pH is measured daily.

In the discussion, the authors highlight the issues with empirical data collection, and the requirement of the field to coalesce around a set of standardised methodologies. However, in their own approach, the methodology employed by the authors is certainly far from standardised, optimised or appropriate for the way the data is laid out.

Undoubtedly this experimental design is a result of constraining factors outside the author’s control (e.g. the requirement to run separate treatments over subsequent years). This alone does not prevent the data being publishable (provided the major issue with the replication can be better explained as shown to be a miscommunication of the real level of replication, which must be far greater). If the data was presented as 3 discrete components/separate experiments (e.g. impact of sex on pCO2 impacts at 17 degrees/ impact of pCO2 on growth at 24 degrees/ impact of growth at 28 degrees), it could add valuable data (albeit far less elegant than the intended aim of this study as initially presented) to the field.

Reviewer #2: I find that the questions asked by the authors to be of contemporary scientific interest and the model species used to be suitable for the questions raised. However, I think that there are certain drawbacks that would require major revisions in the manuscript regarding clarification/reanalyses when it comes to the experimental design and revision of the results that are currently presented in a confusing manner. Please find my specific comments to be detailed below.

Major:

Title: Please rephrase this to sentence as it is currently quite vague.

Line 50: Dated reference, based on what scenario? Latest IPCC projections vary.

Line 52-54: This comes across very dire. Please present negative, neutral and positive effects in the introduction as perhaps see the latest from Jutfelt group (DOI: 10.1038/s41586-019-1903-y).

Line 115: How many of these tanks per treatment? Line 169 suggests to me that everything may be pseudo replicated. One of my main concerns throughout the manuscript is that the trials should be the replicate and within each trial, organisms face the same water (static tanks, correct?) and are therefore pseudo replicates. If I got this wrong, I recommend making this clearer in the methods.

Line 137: Reference to how long these trials lasted either in table or text here please.

Line 195: Discards not sexed?

Line 196: What were the mortality rates?

Line 214: This needs to be tested.

Lines 216-217: Were checks made for baseline differences between trials?

Lines 225-227: Please revise, confusing.

Line 225: Line 217 suggests this was checked at 24C for trial three, so this is confusing. If literature suggests effects occur above 20C, then why test 17C? Why is the interaction of higher temperature and pH not interesting to the authors?

Line 230: Seeing the interesting results on Fultons k, I am unsure why wW isn’t included.

Line 242: The same individual was not compared over time, so a pairwise t-test may not be appropriate.

Line 247: Prefer a formal stats test for these.

Lines 250-256, Line 267: Trials may have to replicate here. They are the source of biological variability in experimental design.

Line 270: Ok, but what about sex specific effects?

Line 274-275: Confusing with respect to line 271.

Line 298: Is this essentially the control?

Line 307: “co2 effect sizes” which direction?

Line 307-308: Or there is simply no effect?

Line 309: Why not no evidence?

Line 315: Ok, but that life history stage is not tested here either.

Line 319: on what aspects of reprodcutove success?

Line 322: Not sure if the process of sex determination is the one investigated here.

Line 323: Why only 2000 in line 305 then?

Line 327-328: Again, to me this evidence suggesting that there are different baslines across trials since they are likely the true replicates. Also, this sentence is quite confusing. Either discuss the 80’s data or don’t but brushing over what is presents seems pointless.

Lines 322-334: Why not discuss the temperature differences in sex ratio here?

Line 329: What are these sex ratios?

Line 349-350: To me, it would be interesting if this is in the gonads or where specifically?

Line 354: What size is the large size and how much higher survival? How does it compare to your data?

Line 355-356: Does this account for higher fultons k observed at higher pCO2?

Line 358-359: But what if smaller fish produce better quality eggs?

Note: Authors lead discussion along a premise of individuals becoming smaller = negative consequences. I’d like to see some discussion of if there can be a selection towards smakker sied individuals with higher fultons k and what that might imply.

Lines 362-363: What concentrations of CO2, what temperature levels? Ie are they comparable? This is too brief, please discuss conflicting data.

Lines 370-378: These line are repetitive to text in methods/introduction.

Lines 386-388: without the test, this statement is speculative. Remove.

Lines 394-396: OK, but what about trials 3 and 4?

Line 397: How can this be environmentally explained?

Lines 401-403: Obvious, remove. With this in mind, authors should have applied these practices to lines 395-396.

Minor:

Abstract: Please go through the text to standardise text for species names (italics) and pCO2 as in the main manuscript.

Line 38: Not clear what the control temperature treatment is.

Line 57: What latitudes/environments?

Line 67: Well it’s not just likely, in certain cases, it has been demonstrated. See Mittermayer et al 2019 Scientific Reports for one example.

Line 68: too small to detect but what about theoretical calculations?

Lines 71-73: Again, see recent paper from the Jutfelt group.

Line 88: How strong is this relationship?

Line 93: “more male biased” This is too vague, please be more specific.

Line 93: “low pH conditions” Please provide the range of these conditions.

Line 94: Specify what the warm conditions are.

Line 138: How long were samples stored in these conditions and how?

Line 157: How were they separated by sex?

Line 157: What are these low densities? Temperature is a wide range..

Line 162: At what ratio? When was fertilisation estimated?

Line 176: Specify PAR

Line 183: Concentration of shrimp nauplii and small rations?

Line 185: Time period, concentrations?

Lines 193-194: When, how much?

Line 230: Change intercorrelated to correlated.

Line 250: Body size date?

Line 257: “long term exposure” What is this period exactly?

Lines 261-262: In which treatments?

Line 291: Grammar.

Lines 325-326: Mechanism not studied here, too speculative.

Line 352-353: Remove.

Line 382: What life stage?

Line 409: Change determination to ratios. Also, check grammar of following sentence.

Line 673, Table 1: Why the different number of days within the 24C treatment? Why the different number of tanks?

Line 682, Table 2: Please provide HCO3- data.

Line 691, Table 3: Can ages for tables 1 and 3 be given consistently? Trial 3 looks like it might have significant effects but is not discussed in line 250-256?

Line 726, Figure 1: dotted lines not needed as the trend is not studied. Also, for panel 1 (17C), seems higher within each trial. Maybe paired tests here with the trial as the replicate and check for indepedant temperature effects later?

Figure 2: What about trial 3 in table 3?

Figure 3: Remove dotted lines since same individual TL is not being tracked.

Figure 4: Panel 3, bottom. What is trial 5?

6. PLOS authors have the option to publish the peer review history of their article (what does this mean?). If published, this will include your full peer review and any attached files.

Reviewer #1: No

Reviewer #2: No

---

## [Author Response · Author response to Decision Letter 0]

30 Apr 2020

To address the additional requirements listed in the decision letter, we have thoroughly checked that our manuscript meets PLOS ONE’s style requirements and have added captions for Supporting Information files at the end of the manuscript. No additional permits were required for the collection of wild Menidia menidia or for access to our collection site. Lastly, a portion of the samples analyzed in this study was resampled from the peer-reviewed manuscript: Murray CS, Fuiman LA, Baumann H. Consequences of elevated CO2 exposure across multiple life stages in a coastal forage fish. ICES Journal of Marine Science. 2017;74(4):1051-61. doi: 10.1093/icesjms/fsw179. This does not constitute dual publication because these resampled fish were dissected for sex identification, which enabled the analysis of sex-specific effects of high pCO2. Sex-specific effects were not addressed in Murray et al. (2017).

---

## [Decision Letter · Decision Letter 1]

24 Jun 2020

Are long-term growth responses to elevated pCO2 sex-specific in fish?

PONE-D-19-33606R1

Dear Dr. Murray,

We’re pleased to inform you that your manuscript has been judged scientifically suitable for publication and will be formally accepted for publication once it meets all outstanding technical requirements.

Kind regards,

Frank Melzner

Academic Editor

PLOS ONE

Additional Editor Comments (optional):

Reviewers' comments:

Reviewer's Responses to Questions

**Comments to the Author**

1. If the authors have adequately addressed your comments raised in a previous round of review and you feel that this manuscript is now acceptable for publication, you may indicate that here to bypass the “Comments to the Author” section, enter your conflict of interest statement in the “Confidential to Editor” section, and submit your "Accept" recommendation.

Reviewer #1: All comments have been addressed

Reviewer #2: All comments have been addressed

2. Is the manuscript technically sound, and do the data support the conclusions?

Reviewer #1: Yes

Reviewer #2: Yes

3. Has the statistical analysis been performed appropriately and rigorously? 

Reviewer #1: Yes

Reviewer #2: Yes

4. Have the authors made all data underlying the findings in their manuscript fully available?

Reviewer #1: Yes

Reviewer #2: Yes

5. Is the manuscript presented in an intelligible fashion and written in standard English?

Reviewer #1: Yes

Reviewer #2: Yes

6. Review Comments to the Author

Reviewer #1: In re-reviewing the manuscript 'Are long-term growth responses to elevated pCO2 sex-specific in fish?' it is clear the authors have undertaken a significant amount of work to address the comments made by both reviewers. Undoubtedly, the manuscript on first submission was topical, interesting and had significant potential merit for the scientific community. However, as also highlighted there were a number of significant issues that needed addressing before the paper could be accepted for publication.

The authors have done a really commendable job in taking on board all the constructive comments made by both reviewers, and in revising have thus significantly strengthened quality of the manuscript. I particularly liked the approach used with the shift analysis. This presented an elegant way of assessing the response profiles of the treatment groups, rather than focussing on an average response - which fails to capture inter-individual variability across the population demographic.

Whilst it is clear it was not possible to address all reviewer comments - indeed the concern over psuedo-replication in trial 4 remains and is valid, the authors concede this was due to a limitation in space availability for this experiment. The authors outline this explicitly in the methods and omit LMM on this trial for this reason. Therefore I do not feel this issue is sufficient to preclude publication. All experiments are on some level an abstraction of reality, and need interpreting in light of their limitations. The revised manuscript clearly presents what experiments have been undertaken, and more importantly the caution that needs to be taken when interpreting the results. I therefore feel it will make a valuable addition to the scientific literature, and now recommend it be published in PLOS One.

Reviewer #2: I am happy to see the revisions implemented to the manuscript particularly in relation to the statistical methods applied and details now provided for the level of replication. Additionally, the revisions requested to the introduction and results were adequately addressed. Typically, I’d still like to see calculated HCO3- values and I believe authors can overcome page formatting issues for the table.

Finally, I have some minor suggestions for the revised file to the authors:

Abstract, Line 32: Grammar of ‘model forage fish..’

Introduction, Line 67: Technically only a fraction is also studied here (1/3rd). I guess authors mean a small fraction?

Line 44 in the abstract (“expectedly”) conflicts with lines 88-90 of the introduction.

Line 164: How were the separated by sex?

7. PLOS authors have the option to publish the peer review history of their article (what does this mean?). If published, this will include your full peer review and any attached files.

Reviewer #1: No

Reviewer #2: No

---

## [Editor Report · Acceptance letter]

6 Jul 2020

PONE-D-19-33606R1 

Are long-term growth responses to elevated pCO2 sex-specific in fish? 

Dear Dr. Murray:

I'm pleased to inform you that your manuscript has been deemed suitable for publication in PLOS ONE. Congratulations! Your manuscript is now with our production department. 

Kind regards, 

on behalf of

Dr. Frank Melzner 

Academic Editor

PLOS ONE